# A Study on the Recovery of Head and the Total Dissolved Solids (TDS) from Long-Term Pressure Depressions in Low Permeable Coastal Aquifers

**Huali Chen [1], Guoping Ding [2], Cheng Hu [2], Eungyu Park [3],\*, Yeongkyoo Kim [3] and Jina Jeong [3]**

[1] School of Environmental Science and Engineering, Zhejiang Gongshang University, Hangzhou 310018, China; hualichen57@126.com

[2] Department of Water Resources and Hydrogeology, School of Environmental Studies, China University of Geosciences at Wuhan, Wuhan 430074, China; gpding@gmail.com (G.D.); hu_cheng@cug.edu.cn (C.H.)

[3] Department of Geology, Kyungpook National University, Daegu, 702-701, Korea; ygkim@knu.ac.kr (Y.K.); jeong.j@knu.ac.kr (J.J.)

\* Correspondence: egpark@knu.ac.kr; Tel.: +82 53 950-5362

**Abstract:** Studies on the recovery of head and total dissolved solids (TDS) in a coastal aquifer system from long-term pressure depressions because of groundwater abstraction (e.g., pumping) is essential for freshwater protection and seawater-intrusion prevention in coastal areas. A 2D numerical model is applied in this paper to investigate the recovery of head and TDS in terms of long-term behavior considering low permeability media. The spatial behavior of the transition zone (TZ), which was chosen as an indicator, was studied in depth with respect to the participant hydraulic and solute-transport characteristics of the aquifer. The sensitivity of the TZ to different aquifer parameters was evaluated. The hydraulic conductivity and rainfall recharge are the two most sensitive factors that affect the location of the TZ in homogeneous cases, and the spatial structure of the hydraulic conductivity field, namely, the correlation length and variance, largely influences the sensitivity of the TZ. The required time for the complete recovery of head in the heterogeneous cases is much shorter than that in the homogeneous cases, but the TDS recovery takes much more time. When the recovery of head is 90%, low porosity and large specific storage play an important role in the location of the TZ compared to other parameters, except for the hydraulic conductivity and recharge rate. The results of this study are meaningful for coastal-aquifer management and may be instructive in the restoration of coastal areas that have experienced seawater intrusion because of the long-term overexploitation of fresh groundwater.

**Keywords:** recovery of head & TDS; coastal aquifers; transition zone between freshwater and saltwater; centroid; sensitivity analysis

## 1. Introduction

Many of the world's populations are sustained by coastal aquifers. As of the 1990s, coastal populations comprised 53% of the entire population of the USA [1]. In many developing and developed countries, people often crowd cities that are situated near coastal areas. The preservation of coastal aquifer systems is important because groundwater is frequently the sole source of freshwater. Such large populations, however, usually impose great pressure on coastal aquifer systems, and the inevitable demand on water consequently deteriorates the quality and quantity of coastal groundwater resources. Typical human activity with regards to coastal aquifer systems entails the extraction of freshwater from the subsurface for drinking, domestic, industrial, agricultural, and fishery purposes.

Freshwater extraction between freshwater rates often exceed natural replenishment rates, and therefore interrupt the balance and seawater [2–7]. Thus, seawater moves landward, which is called seawater intrusion, and the portion of the aquifer that used to be occupied by freshwater is replaced by saltwater. Saltwater intrusion may significantly impair the groundwater resources in a coastal aquifer, affecting the relevant population. In the last several decades, many developed countries have experienced serious seawater intrusion problems. One solution is to stop pumping from the aquifer to allow the recovery of the balance between freshwater and seawater. Many research projects have been conducted to understand the characteristics of the recovery process. Bear et al. simulated seawater intrusions and recoveries in response to pumping-well activation and deactivation, and the parameters that were used in the model were based on the flow and transport conditions in the coastal aquifer of Israel [8]. The total simulation time was 50 years, and these authors noted that the entire system would require much more time to completely recover to the initial conditions without pumping. However, neither the recovery of head nor the effect of the parameters on the recovery of salinity was discussed. Zhou et al. studied the upconing and decay of salinity beneath a pumping well over an interface zone. The upconing and decay of the TDS were simulated by a well that pumped at a fixed rate of 100 $m^3$/h, and shut-off occurred when the normalized mass fraction of salt in the pumped water reached 2% [4]. The total simulation time was 80 years for the brine case. However, no regional flow was observed, and the results did not represent the head and recovery of TDS processes in a real coastal aquifer system. Miust and Voss simulated the movement of the freshwater-saltwater transition zones in real coastal aquifers by injecting freshwater into aquifers for the purpose of freshwater storage and recovery by extraction [9]. This simulation covered a 10,000-year period of increasing sea level, and TDS movement was still in progress by the end of the simulation. Although the movement of the freshwater-saltwater transition zone was studied with respect to freshwater injection and extraction, this study focused on the efficiency of aquifer storage and recovery. Furthermore, the model utilized freshwater injection, which is not always the case in reality.

The decay in increased salinity that is induced by pumping is expected to take a long time, and the characteristics of the recovery process greatly depend on the flow and transport characteristics but have not been systematically addressed in the literature. In this paper, the recovery process of head and the TDS in terms of long-term behavior after pumping is studied by using a 2D numerical model considering low permeability media. The location of the freshwater-saltwater transition zone is taken as an indicator to study the recovery process. The spatial behavior of the transition zone is studied in depth by using a numerical approach with respect to the participant hydraulic and solute transport characteristics, including the hydraulic conductivity, anisotropy of the hydraulic conductivity, storativity, porosity, dispersivity, and recharge rate. The heterogeneity of the hydraulic conductivity is also considered to study the effect of the spatial structures of the hydraulic conductivity distribution. The sensitivity of the recession of the transition zone to different parameters is evaluated. The results of this study can provide useful information for the control and management of recovery processes in a real coastal aquifer to establish necessary countermeasures.

## 2. Methodology

In coastal aquifer systems, the pressure balance between freshwater and seawater may be measured by the location of the interface. Many aquifer properties and boundary conditions influence the location of the interface. In this study, the coastal aquifer is assumed to have reached its steady state in terms of head and TDS at the initial time after a long period of pumping. More detailed procedures for preparing the initial conditions of the examined cases are described in Section 3: "Model Setup". From the initial state, the pumping well is assumed to cease its operation. The TDS and the pressure of the system recover to the original conditions before the start of pumping. A one-hundred-year transient-recovery period is observed by using numerical simulations. Theoretically, a line sink in two-dimensional space does not reflect well pumping in three dimensions, where the flow around a wellbore is essentially radial. However, the main objective of this study is to investigate the recovery capability of coastal aquifers

from long-term pressure depressions and TDS modifications, and the depressions and modifications do not necessarily resemble reality.

As stated above, the variable conditions of an aquifer system, including the aquifer parameters, may have different effects on the location and movement pattern of the interface. To evaluate the sensitivity of the location and movement against the aquifer parameters, twelve different cases were designed (Table 1); the principles of parameter selection are described in Section 4.1.1: "Case Design". The actual values from a coastal aquifer that consists of fractured granite with a subsurface disposal facility of low-to intermediate-level radioactive waste (Gyeongju in South Korea) were referenced when determining the parameters [10].

**Table 1.** Parameters that were used for the homogeneous case.

| | Hydraulic Conductivity (m/s) | Anisotropic Ratio | Recharge (m/day) | Porosity | Specific Storage (m$^{-1}$) | Longitudinal Dispersivity (m) |
|---|---|---|---|---|---|---|
| Base case | $5.0 \times 10^{-7}$ | 1 | $3.64 \times 10^{-4}$ | 0.0035 | $1.0 \times 10^{-4}$ | 100 |
| Case 1 | $5.0 \times 10^{-7}$ | 1 | $3.64 \times 10^{-4}$ | 0.0035 | $1.0 \times 10^{-4}$ | 66.7 |
| Case 2 | $5.0 \times 10^{-7}$ | 1 | $3.64 \times 10^{-4}$ | 0.0035 | $1.0 \times 10^{-4}$ | 200 |
| Case 3 | $2.5 \times 10^{-7}$ | 1 | $3.64 \times 10^{-4}$ | 0.0035 | $1.0 \times 10^{-4}$ | 100 |
| Case 4 | $6.0 \times 10^{-7}$ (Hmin = −15 m) | 1 | $3.64 \times 10^{-4}$ | 0.0035 | $1.0 \times 10^{-4}$ | 100 |
| Case 5 | $5.0 \times 10^{-7}$ | 0.5 | $3.64 \times 10^{-4}$ | 0.0035 | $1.0 \times 10^{-4}$ | 100 |
| Case 6 | $5.0 \times 10^{-7}$ | 2 | $3.64 \times 10^{-4}$ | 0.0035 | $1.0 \times 10^{-4}$ | 100 |
| Case 7 | $5.0 \times 10^{-7}$ | 1 | $3.64 \times 10^{-4}$ | 0.00175 | $1.0 \times 10^{-4}$ | 100 |
| Case 8 | $5.0 \times 10^{-7}$ | 1 | $3.64 \times 10^{-4}$ | 0.0070 | $1.0 \times 10^{-4}$ | 100 |
| Case 9 | $5.0 \times 10^{-7}$ | 1 | $3.03 \times 10^{-4}$ (Hmin = −15 m) | 0.0035 | $1.0 \times 10^{-4}$ | 100 |
| Case 10 | $5.0 \times 10^{-7}$ | 1 | $5.46 \times 10^{-4}$ | 0.0035 | $1.0 \times 10^{-4}$ | 100 |
| Case 11 | $5.0 \times 10^{-7}$ | 1 | $3.64 \times 10^{-4}$ | 0.0035 | $5.0 \times 10^{-5}$ | 100 |
| Case 12 | $5.0 \times 10^{-7}$ | 1 | $3.64 \times 10^{-4}$ | 0.0035 | $2.0 \times 10^{-4}$ | 100 |

## 2.1. Spatial Measurements for the TZ

In this study, the main consideration was the TZ's distribution and transient movement. Previously, the location of the interface between freshwater and saltwater has been commonly used for the spatial measurement of the TZ [11,12]. This measurement is clear in graphic form when only a steady-state distribution of the interface is considered. For the transient movement of the interface, however, the interface curve is not sufficient for detailed observations.

The spatial distribution of the TZ over time can be more effectively represented by spatial-moment analysis. Spatial-moment estimation has long been used to define the characteristics of a contaminant plume versus time [13–15]. In this study, the TZ was treated as a mobile plume and the first spatial moment (i.e., the centroid of mass) was applied to characterize the movement of the TZ with time. The coordinates of the centroid were defined by the following equation [15]:

$$u_i = \frac{\int_\Omega nC(\mathbf{x},t)x_i d\Omega}{\int_\Omega nC(\mathbf{x},t)d\Omega} \qquad i = 1,2,\ldots,D \tag{1}$$

where $u_i$ is the centroid coordinate in the $i$th direction, $D$ is the spatial dimension, $n$ is the effective porosity, and $C(\mathbf{x},t)$ is the TDS at location x($x_1,x_2,\ldots,x_D$) and time $t$. $\Omega$ is the domain that is occupied by the TZ.

## 2.2. Centroid-Coordinate Rescaling

Different parameters may cause large displacements of TZ centroids. We must rescale the coordinates of the centroids for different cases so that these values can have similar ranges

and so that we can quantitatively measure the displacement of the centroid with different parameters and different times. A common rescaling formula is defined as follows [16]:

$$a_i = \frac{(A_i - A_{\min})}{(A_{\max} - A_{\min})} \qquad i = 1, 2, \ldots, D \tag{2}$$

where $a_i$ is the rescaled coordinate value in the $i$th direction; $D$ is the spatial dimension; $A_i$ is the original coordinate value in the $i$th direction; and $A_{\min}$ and $A_{\max}$ are the minimum and maximum values of $A_i$, respectively.

### 2.3. Unconditional LogK-Field Generation

Both homogeneous and heterogeneous aquifer properties were considered in this study. For the heterogeneous aquifer, the hydraulic conductivity was assumed to be log-normally distributed with specified horizontal and vertical correlation lengths [14,17]. The hydraulic conductivity value that was used in the homogeneous cases was set as the mean of that for the heterogeneous cases. The variance of the hydraulic-conductivity field was carefully determined not to cause a convergence problem in the numerical model. A sequential Gaussian simulation (SGSIM) model was used to generate the heterogeneous random distribution [16]. The parameters that were used for the unconditional simulation are tabulated in Table 2.

**Table 2.** Parameters that were used for the heterogeneous case.

|  | Mean of lg(K) (m/s) | $\lambda_{\max}$ (m) | $\lambda_{\min}$ (m) | Variance | $H_{\min}$ (m) |
|---|---|---|---|---|---|
| New base case | −6.3 |  |  | 0 | 0 |
| Case 1 | −6.3 | 60.0 | 20.0 | 0.1 | 0.0 |
| Case 2 | −6.3 | 90.0 | 30.0 | 0.1 | 0.0 |
| Case 3 | −6.3 | 120.0 | 40.0 | 0.1 | 0.0 |
| Case 4 | −6.3 | 60.0 | 20.0 | 0.2 | 0.0 |
| Case 5 | −6.3 | 90.0 | 30.0 | 0.2 | 0.0 |
| Case 6 | −6.3 | 120.0 | 40.0 | 0.2 | 0.0 |
| Case 7 | −6.3 | 60.0 | 60.0 | 0.1 | 0.0 |
| Case 8 | −6.3 | 90.0 | 90.0 | 0.1 | 0.0 |
| Case 9 | −6.3 | 120.0 | 120.0 | 0.1 | 0.0 |
| Case 10 | −6.3 | 60.0 | 60.0 | 0.2 | 0.0 |
| Case 11 | −6.3 | 90.0 | 90.0 | 0.2 | 0.0 |
| Case 12 | −6.3 | 120.0 | 120.0 | 0.2 | 0.0 |

## 3. Model Setup

### 3.1. Conceptual Model

A hypothetical two-dimensional (2D) coupled density-driven flow and solute-transport model that referenced the geometry and hydraulic properties of a coastal aquifer in Gyeongju, South Korea [10] was considered in this study (Figure 1). The conceptual model domain is shown in Figure 1. The domain was 2532.1 m in width and 1519.3 m in height and contained two components: a land portion, which had a width of 2153.8 m, and a sea portion, which had a width of 378.3 m. The sea level was set to 0 m. The topography of the land portion was higher on the left side of the domain and became lower on the right side with a mean gradient of 0.21. A constant recharge rate was assumed for the top boundary that was exposed to the atmosphere. The left-hand side and bottom of the domain were set as no-flow and no-mass flux boundaries. The right-hand side was initially set as a constant head with constant concentration in terms of TDS where flow was inward and a zero-concentration gradient where flow was outward. Similar boundary conditions were set at the top side of the sea portion. Density effects on the hydraulic head were considered, and the hydraulic head was expressed in terms of freshwater. The hypothetical pumping well is shown in Figure 1. The distance of the well from the shoreline

was 353.8 m, and the penetration depth of the well was 90 m. The ground level of the pumping well in the model was 66.6 m.

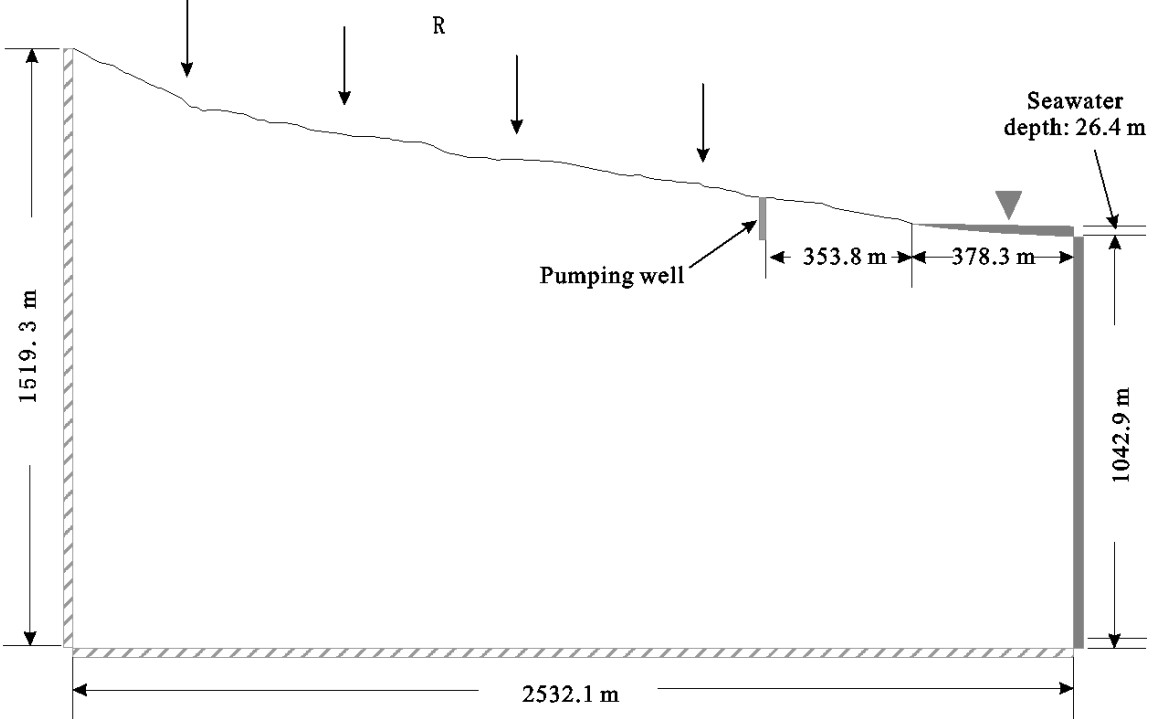

**Figure 1.** Map of the conceptual model that was used in the hypothetical 2D study.

### 3.2. Spatial Discretization

In this study, FEFLOW® [18] was used for the density-driven flow and TDS-transport simulations. A super element was created in FEFLOW® based on the above conceptual model, and the 2D triangular mesh was automatically generated and optimized. To improve the spatial resolution of the transition zone, a finer mesh was applied near the pumping well and coastline. The number of nodes in the final mesh was 11,065, and the number of elements was 21,733.

### 3.3. Basic Model Parameters

The density of saltwater is 1.025 g/cm$^3$, and the density ratio is 0.025. The molecular diffusion coefficient is 1 m$^2$/s. The TDS of saltwater is 35,000 mg/L, and the TDS of freshwater is 0 mg/L. Decay and sorption were not considered in the model. The values for hydraulic parameters such as the hydraulic conductivity, anisotropic ratio of the vertical hydraulic conductivity to the horizontal conductivity ($K_v/K_h$), porosity, specific storage, and dispersivity will be discussed in the following sensitivity-analysis section.

### 3.4. Boundary Conditions and Initial Conditions

A constant precipitation rate of 1.33 m/year was assumed for the top surface of the land portion, and the infiltration out of the precipitation will be discussed in the sensitivity-analysis section. As already stated in the conceptual model, the left and bottom boundaries were set as no-flow and no-mass flux boundaries. For the right-hand side and top surface of the sea portion, a constant head was assigned to the corresponding nodes while considering the density effects. The TDS concentrations of these nodes were 35,000 mg/L and the conditions were constrained such that no flow efflux occurred. If seaward flux occurred from these nodes, the TDS concentrations were computed by the model. For all the modeled cases, steady-state models with continuous pumping were first run with the corresponding input

parameters to derive the distributions of the head and TDS, which were then used as the initial conditions for the entire internal model domains of the cases.

## 4. Simulation Results and Discussion

### 4.1. Homogeneous Case Studies

#### 4.1.1. Case Designs

For flow and transport simulations, technically improper hydraulic parameter values, such as extremely high or low hydraulic conductivity (K), may cause unreasonable results, especially when considering effective drawdown during pumping and the following recovery process (e.g., no effective drawdown for excessively high K values and unrealistically significant drawdown for excessively low K values). Therefore, a trial-and-error method was used to delineate 'reasonable values' for the selected aquifer settings in Figure 1 and set the values as a base case. The values that were delineated from a coastal aquifer in Gyeongju, South Korea were referenced for the ranges of the actual hydraulic parameters [10]. Other cases for sensitivity analyses were designed by changing the value of a different target parameter while the other parameters were fixed.

The parameters that were used for the base case are listed in Table 1. In the base case, the hydraulic conductivity was $5.0 \times 10^{-7}$ m/s, the anisotropic ratio was 1, the ratio of the infiltration rate to the precipitation rate was 0.1, the porosity was 0.0035, the longitudinal dispersivity was 100 m, the ratio of the longitudinal to transverse dispersivity was 10, and the specific storage was $1.0 \times 10^{-4}$ m$^{-1}$.

Other cases were designed based on the base case. For each case, only one parameter was changed; all the other parameters were kept the same as those of the base case. The value of each parameter was 2.0 times larger or smaller than that of the base case. For example, the value of the hydraulic conductivity in the base case was $5.0 \times 10^{-7}$ m/s, while this value in case 3 was changed to $5.0 \times 10^{-7}/2 = 2.5 \times 10^{-7}$ m/s. However, some exceptions occurred in cases 4 and 9 because of the convergence problem. For these cases, a factor of 1.5 or smaller was applied instead of 2.0 because of numerical instability, and the target minimum head that was caused by pumping at the pumping location, $H_{min}$, was changed from −20 to −15 m (Table 1).

#### 4.1.2. Base Case Study

In the base case study, a steady-state model was first set up to derive the steady distribution of the head and TDS in the model domain. The pumping rate for the hypothetical pumping well was determined inversely by setting the target minimum head as a specified head along the pumping well. With this derived pumping rate, the initial conditions of the head and TDS for the cases were derived and the recovery processes of the head and TDS were simulated by running the transient model. The recovery of head was measured by the time that was required for a certain percentage of head recovery, which is defined as the ratio of the head at the desired time to that in the steady state without pumping at the pumping well. Figure 2 shows the time that was required for different percentages of head recovery for the base case together with those for the other twelve cases. We found that 80% head recovery was achieved within a relatively short period for all thirteen cases (e.g., 148 day for the base case), and 90% head recovery was also achieved quickly compared to the total simulation period of 100 years. The maximum required time for 90% head recovery was 4078 day in case 6. Figure 2 shows that less than 10 years was required for 90% head recovery in most cases. The complete recovery of head may take longer, but reached 100% before the end of the simulations.

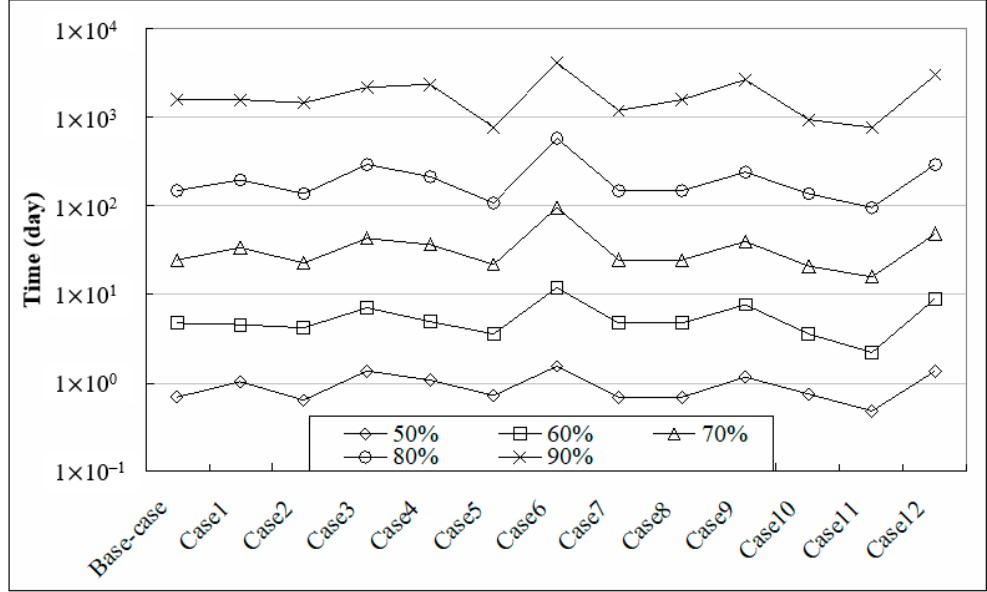

**Figure 2.** Required time for different percentages of head recovery.

Figure 3 shows the contour lines of 1000 mg/L TDS as the base case when the percentage of head recovery was 0%, 50%, 60%, 70%, 80%, and 90% alongside the end of the simulation. The contour line for the steady state without pumping is also included. The minimum TDS concentration of the TZ is 1000 mg/L [19], so these 1000-mg/L TDS contours serve as indicators of the location of the interface between freshwater and saltwater. According to Figure 3, the TDS recovery was completely different from the head recovery. When the head recovery reached 80%, the interface did not apparently change. Even when the head recovery reached 90%, the upper portion of the interface showed a small recession while the lower portion had no obvious movement. Significant recovery of TDS occurs when the percentage of head recovery is larger than 90%. At the end of the simulation, the 1000-mg/L TDS contour was still detached from the line for the steady state without pumping, suggesting that the recovery of TDS had not finished even after 100 years of simulation time. A much longer time may be required to reach the complete recovery of TDS.

The 1000-mg/L TDS contours in Figure 3 suggest that the recovery of TDS was fast in the upper aquifer, especially near the shore area, and that the recovery process was quite slow in the deep portion of the aquifer. These results occurred because the shallower aquifer near the shore area is the main submarine groundwater-discharge area, where changes in groundwater flow because of pumping are dominant. In the deep portion of the aquifer, the hydraulic gradient is too low to push the interface back to the original position.

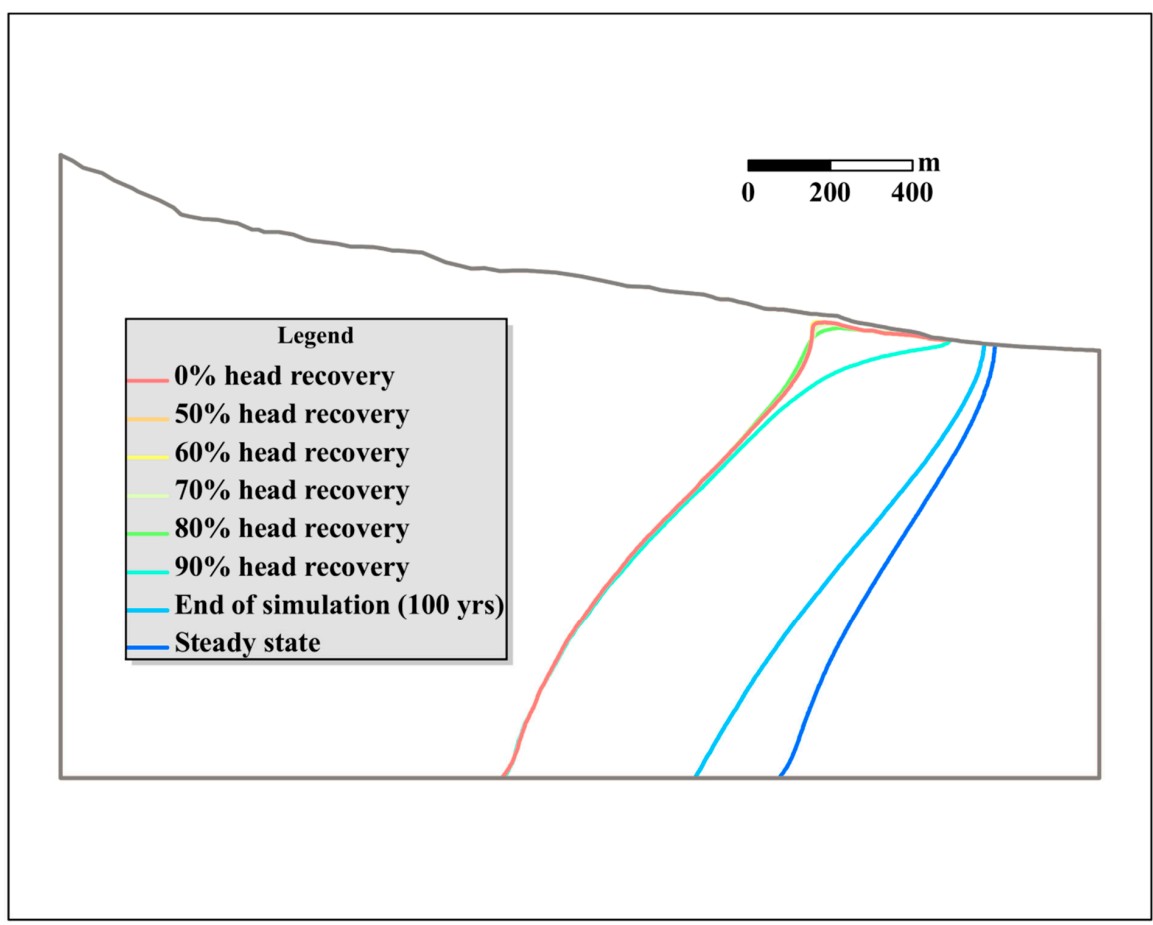

**Figure 3.** Contours of 1000 mg/L TDS for different percentages of head recovery for the base case.

### 4.1.3. Sensitivity Analysis

For the other twelve cases in Table 1, a steady-state model with corresponding parameter settings was set up to derive the steady distribution of the head and TDS, which acted as the initial conditions of the transient model for each case. The same simulation procedures as in the base case study were applied to these twelve cases. The TZ is a diffuse zone, so quantitatively expressing differences in the TZ for the above cases is difficult. Centroids of the TZ at different stages of head recovery for different cases were computed, and the coordinates of these centroids were rescaled for further analysis. The distances of the rescaled centroids for the base case and the other twelve cases at different stages of head recovery were computed. If the distance was large, the displacement of the TZ centroids from the base case was large, suggesting that the TZ was sensitive to the parameter that was adjusted in the corresponding case. Figure 4 shows the distances of the rescaled centroids between the base case and the other twelve cases at different stages of head recovery.

The centroid analyses of the TZ revealed that the TZ was most sensitive to the hydraulic conductivity, corresponding to cases 3 and 4 (Figure 4b), if long-term pumping was conducted in the coastal aquifer with large drawdown, corresponding to 0% head recovery in Figure 4. The second-most sensitive parameter was recharge, corresponding to cases 9 and 10 (Figure 4e). The anisotropy ratio (Figure 4c) played an important role in the movement of the TZ in cases 5 and 6, especially when the vertical hydraulic conductivity was higher than the horizontal conductivity (case 5). The dispersivity (cases 1 and 2 in Figure 4a) was a relatively less sensitive parameter for determining the position and movement of the TZ. The flow system had reached a steady state at this stage, so the effect of the specific storage in cases 11 and 12 (Figure 4f) and the porosity in cases 7 and 8 (Figure 4d) was almost negligible, which was expressed by zero displacement in Figure 4d,f.

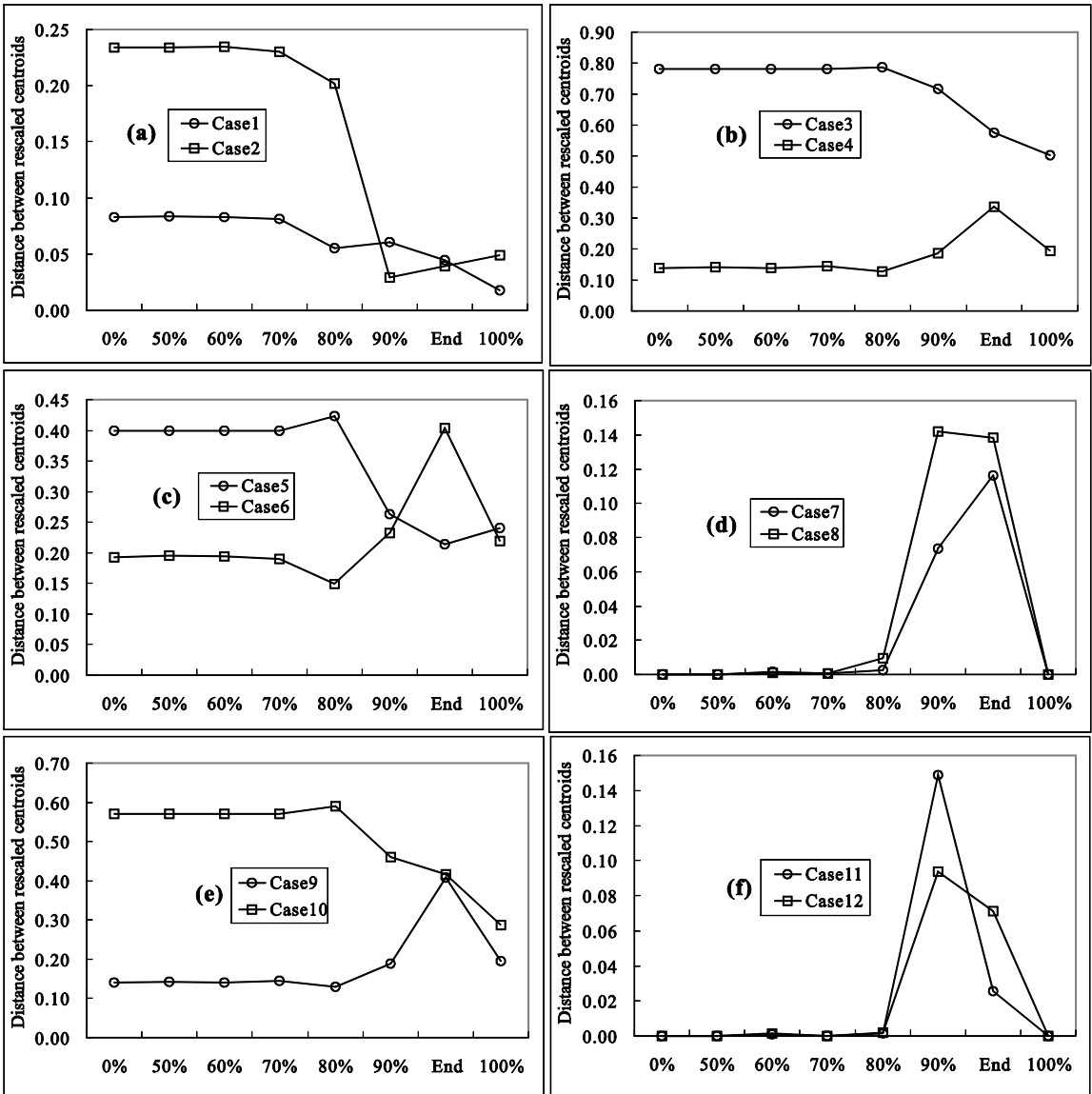

**Figure 4.** Distances of the rescaled centroids between the base case and the other cases at different stages of head recovery in the homogeneous cases.

When the hypothetical well stopped pumping, the recovery process of head and the TDS began. When the head recovery was less than 80%, the displacements did not change. Therefore, the sensitivity of the TZ to different parameters was not obvious. Even when the head recovery was 80%, the displacements of the centroids for all twelve cases were not discernable. At this stage, the displacement of the centroids in cases 7 and 8 (Figure 4d) became nonzero, which suggests that the porosity started to affect the position of the TZ. When the head recovery was larger than or equal to 90%, the displacements of the centroids became large. The hydraulic conductivity, represented in cases 3 and 4 (Figure 4b), was still the most important factor that controlled the TZ's movement. The recharge rate, represented in cases 9 and 10 (Figure 4e), was the second-most important factor, and the anisotropic ratio, represented in cases 5 and 6 (Figure 4c), was the third-most important. At this stage, the sensitivity of the TZ to the porosity in cases 7 and 8 (Figure 4d) and specific storage in cases 11 and 12 (Figure 4f) increased, even becoming larger than that for the dispersivity in cases 1 and 2 (Figure 4a). When the simulation was run for 100 years, the results were similar to when the head recovery was 90%. However, the porosity seemed to be more sensitive than the specific storage and dispersivity. Finally, when the flow and TDS fields reached the new steady

state, the sensitivity rank of the TZ to the parameters did not significantly change, with the exception of the porosity and specific storage, whose contribution to the TZ became zero again.

Figure 4 also suggests that larger dispersivity values were linked to larger displacement when the head was slightly recovered. However, this trend was reversed when the head recovery was 90% and at the end of simulation. This result occurred the flow field approached the normal gravity-driven flow in the domain, especially in areas that were originally occupied by freshwater, when the percentage of head recovery was larger. Larger dispersivity values are linked to quicker TDS recession. When a new balance of freshwater and saltwater was reached, the displacement of the centroids for cases with high dispersivity values became larger, as shown in Figure 4.

At the beginning of recovery, upward flow near the pumping well was dominant because of the depression cone that was caused by long-term pumping. Because of gravity effects, the TZ had difficulty moving inland and towards the depression cone for case 6, where the vertical hydraulic conductivity was higher than the horizontal conductivity. Therefore, the distance in case 5 was larger than that in case 6 for 0% head recovery. When the recovery process started, the TZ receded in both the downward and seaward directions. However, when the percentage of head recovery was small, the recession of the TDS was not obvious. The distances in case 5 remained larger than those in case 6. As head recovery progressed, the downward movement of the TZ became more important and the displacement increased accordingly, although the absolute value was still smaller than that in case 5. At the end of the simulation, the distance in case 6 was larger than that in case 5. When the percentage of head recovery increased and the process finally completed, the TZ finally moved towards the sea. Therefore, when the flow and TDS fields reached a new balance, the horizontal hydraulic conductivity became more important; the distances of the rescaled centroids for the steady state in case 6 were again smaller than those in case 5.

The porosity and specific storage are two special parameters, because they are not involved in the governing equations for a steady-state model. However, Figure 4 demonstrates that these two parameters did affect the TZ's position/movement. When the percentage of head recovery was small, the distances of the rescaled centroids did not change. When the head recovery reached 90%, the distances of the rescaled centroids in cases 7, 8, 11, and 12 (Figure 4d,f) greatly increased. In cases 7 and 8 (Figure 4d), smaller porosity (case 7) and higher seepage velocity expedited the recovery process of the TDS, which explains why the distances in case 7 were smaller than those in case 8. The storage in case 11 was much lower than that in case 12 (Figure 4f), so case 12 took much more time to reach 90% head recovery compared to case 11. Therefore, when the head recovery reached 90% for both cases, the displacement of the TZ in case 12, which had high storage, was reasonably larger than that in case 11. The distances between case 12 and the base case should have been smaller than those between case 11 and the base case. However, when the head recovery was almost completed, the distances between case 12 and the base case at the end of the simulation were larger than those between case 11 and the base case.

The above arguments are based on the displacement of the centroids between the base case and different cases at different stages during head recovery. To understand the recovery of the TDS with time, the distances between different stages of head recovery were computed for the same cases. The accumulated distances from the beginning of recovery to different stages of head recovery are plotted in Figure 5.

According to Figure 5, when the head recovery is less than 80%, the TDS recovery was very small compared to the displacement during groundwater extraction. When 80% of the head was recovered, the TDS recovery was still not obvious. The maximum TDS recovery was observed in case 2, where the accumulated distance comprised approximately 10% of the total TDS recovery. Case 2, in which the dispersivity was larger than in the base case, suggests that dispersion was a dominant process in TDS recovery compared to advection when the percentage of head recovery was low. The process of TDS recovery accelerated when the head recovery reached 90%, and the differences in the amount of TDS recovery for different cases were more peculiar. For example, the accumulated

distances comprised more than 30% of the total TDS recovery in cases 1, 7, and 12. In cases 4 and 9 and the base case, the accumulated distances were between 20% and 30% of the total TDS recovery. In cases 3, 5, 8, 10, and 11, the accumulated TDS recovery was still less than 10%, which suggests that the TDS recovery was still not obvious. Significant TDS recovery occurred in all cases during the period from 90% head recovery to the end of the simulation. In cases 3, 7, and 10, the TZ mostly recovered to its non-pumping steady-state location. However, in cases 4, 6, 8, and 9, the progress of the TZ's recovery was still slow and may have taken much more time to reach 100% TDS recovery. According to the above analyses, most TDS recovery is confined to the period when the head recovery is more than 90%, and complete TDS recovery may take much more time after complete head recovery.

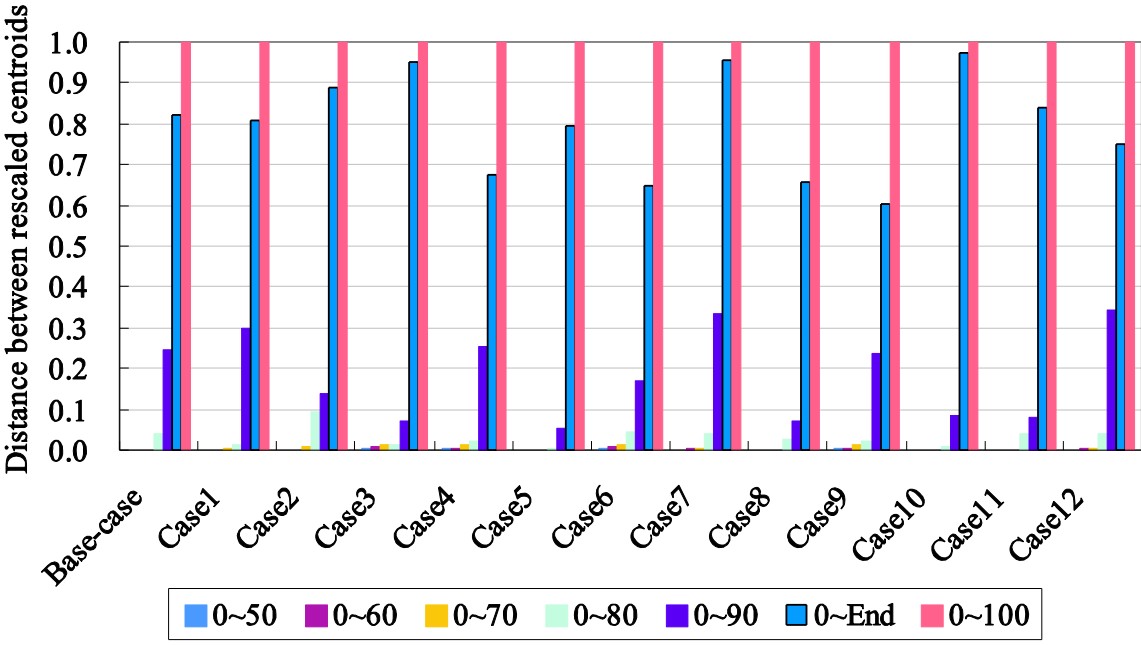

**Figure 5.** Accumulated distances of the rescaled centroids from the beginning of recovery to different stages of head recovery for different homogeneous cases.

*4.2. Heterogeneous Case Studies*

4.2.1. Realization and Case Design

In this section, TZ movement in heterogeneous coastal aquifer systems was also simulated and analyzed as benchmark cases of the previously discussed homogeneous cases, because heterogeneous media are more representative of real coastal aquifer systems, which are typically layered with multiple confining and conducting layers. To simplify the analysis, the log-normal distribution of the hydraulic conductivity, which was determined to be the most sensitive parameter in the previous homogeneous cases, was assumed for the aquifer system. The hydraulic-conductivity value that was used in the homogeneous cases, $5.0 \times 10^{-7}$ m/s, was used as the mean for the generation of the hydraulic-conductivity field. The details of the spatial statistics that were used to generate the heterogeneous random field, including the mean ln$K$, variances of ln$K$, and correlation ranges, are tabulated in Table 2. During the derivation of the variance and correlation length, the value of $H_{min}$ was adjusted, and the values for different cases are listed in Table 2. As shown in this table, both the anisotropy and isotropy of the variogram model were considered by controlling the maximum and minimum correlation lengths (denoted as $\lambda_{max}$ and $\lambda_{min}$, respectively). For comparison, a new base case was designed for the heterogeneous cases. This new base case was actually a homogeneous case with a hydraulic conductivity of $5.0 \times 10^{-7}$ m/s. The other parameters were the same as those in the base case for the homogeneous case study.

Based on the parameters in Table 2, the unconditional SGSIM model was applied to generate a single realization for each case. In the realizations, the mean ln*K* was steered to be −6.3 and the variances, $\sigma_{\ln K}$, were within the range of 0.1–0.2. The data were trimmed if the value was higher than −5.0 or less than −8.0 so as not to cause any numerical complications. The grid size that was used for SGSIM was 10 m × 10 m, and the total number of grids was 41,600.

### 4.2.2. Simulation Results

Similar simulation and analysis procedures as in the homogeneous cases were applied to the current cases in Table 2. Figure 6 shows the required time for differential percentages of head recovery for the new base case and the other 12 cases. We found that 80% head recovery was achieved within 118 day and 90% head recovery was achieved within 1114 day for the new base case, which was the largest among all the cases in Figure 6. The required time for the same percentage of head recovery in the new base case was much shorter than that in the homogeneous cases. This result indicates that the head recovery in the heterogeneous cases was easier than in the homogeneous cases, especially when the isotropic spatial structure of the hydraulic conductivity was considered, as in cases 7–12.

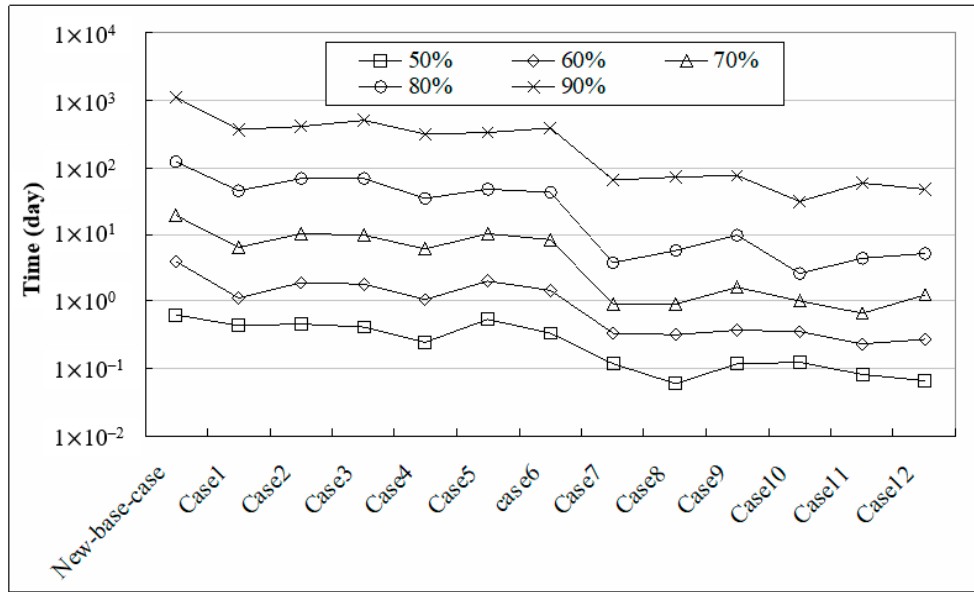

**Figure 6.** Required time for different percentages of head recovery for the heterogeneous cases (note: "New base case" in this figure denotes the new base case).

Figure 7 shows the contour lines of 1000 mg/L TDS for the new base case when the percentage of head recovery was 0%, 50%, 60%, 70%, 80%, and 90% alongside the end of the simulation. The result of the steady-state model without pumping is also presented in Figure 7. The value of $H_{min}$ was 0 m in these cases, so the corresponding pumping rate was much smaller than that in the homogeneous cases, and the upconing effect was not as peculiar as in the previous homogeneous case study. When the head recovery was less than or equal to 80%, no obvious change in the interface was observed. When the head recovery was 90%, the upper portion of the interface receded into the upper portion of the aquifer. However, the interface shape of the lower portion showed no change. The difference between the interface at the end of the 100-year simulation and that in the steady state indicates that no steady-state distribution of the TDS had been reached.

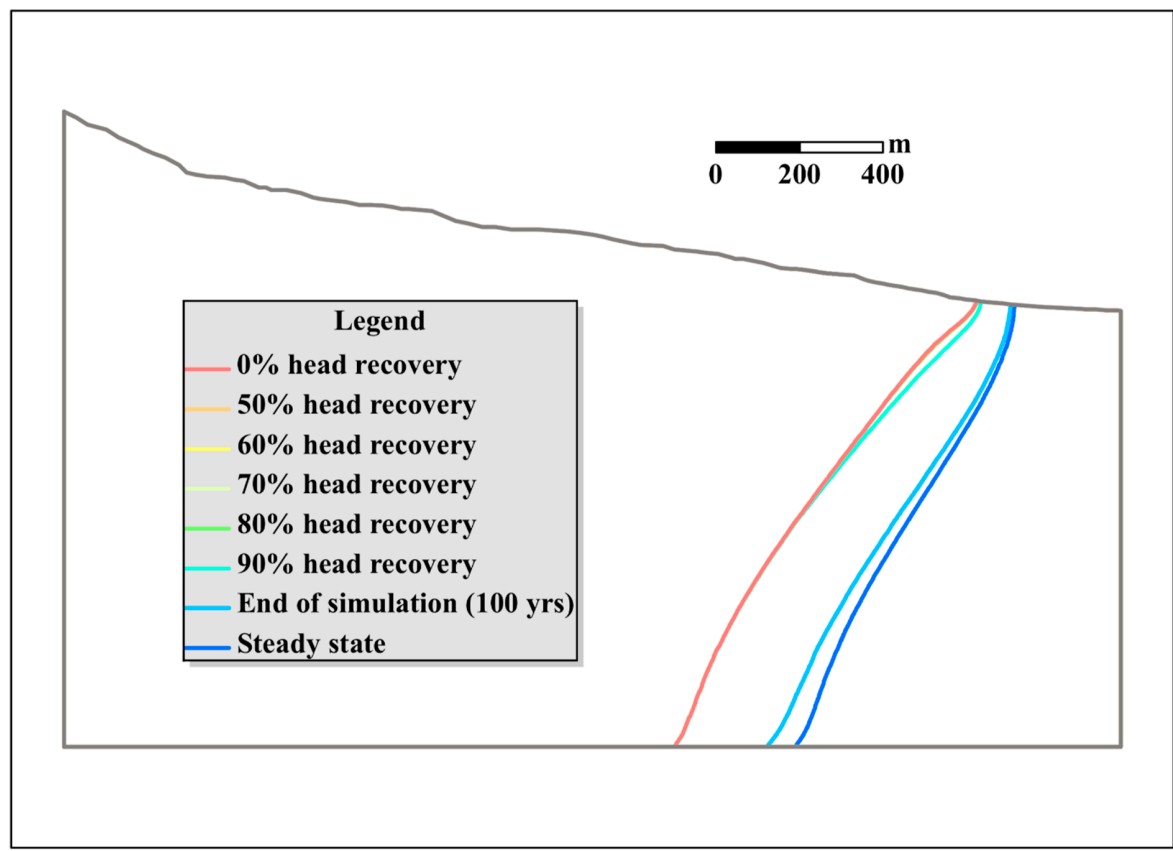

**Figure 7.** Contours of 1000 mg/L TDS for different percentages of head recovery for the new base case.

### 4.2.3. Centroid Movement Considering Aquifer Heterogeneity

Centroids of the TZ were computed at different percentages of head recovery for different cases and plotted in Figure 8. According to this figure, the TZ centroids showed much more extensive landward movement compared to the new base case regardless of the variance or correlation lengths. This finding suggests that a heterogeneous hydraulic-conductivity distribution, which is usually the case in real aquifer systems, accommodated the landward movement of the interface. Therefore, if the heterogeneous hydraulic-conductivity field is simplified as a homogeneous hydraulic-conductivity field for a real aquifer system, the location of the transition zone from the simulation results may be highly underestimated. In Figure 8, the TZ centroids showed upward movement compared to the new base case, especially when the anisotropic spatial structure of the hydraulic-conductivity fields was considered. This observation indicates that heterogeneous cases are advantageous for the upconing of saltwater because of pumping compared to homogeneous cases. Cases that consider the anisotropic spatial structure of the hydraulic-conductivity fields can show further landward and upward movements of the TZ compared to cases that consider the isotropic spatial structure of the hydraulic-conductivity fields.

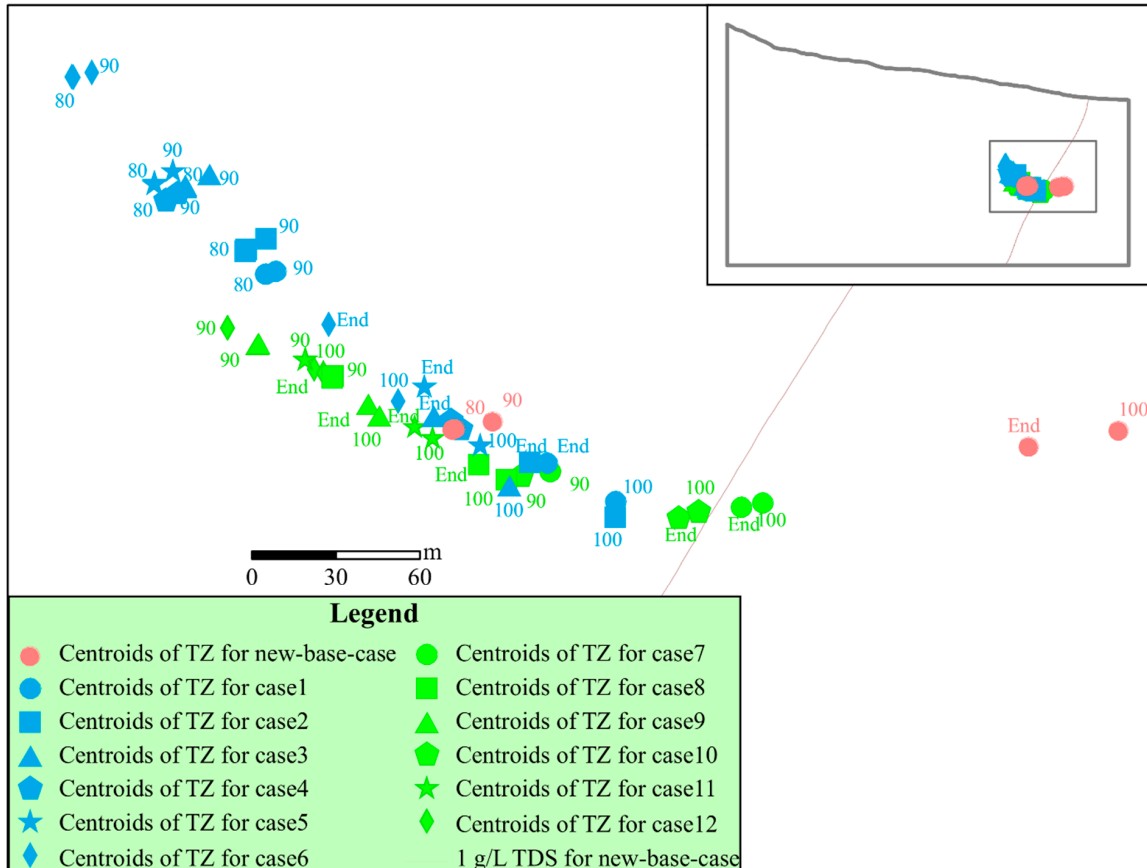

**Figure 8.** Centroid movement for different stages of head recovery in the heterogeneous case study (note: (1) the pink points are for the new base case, the green symbols are for cases 7–12 considering an isotropic spatial structure for the hydraulic conductivity fields, and the blue symbols are for cases 1–6 considering an anisotropic spatial structure for the hydraulic conductivity fields; (2) the labels "80", "90", and "100" denote the percentage of head recovery, and "End" denotes the end of the 100-year simulation).

### 4.2.4. Sensitivity Analysis

The same simulation procedures as in the homogeneous cases were performed on the 12 heterogeneous cases in Table 2. Centroids of the TZ at different stages of head recovery for different cases were computed, and the coordinates of these centroids were rescaled by using Equation (2). The distances of the rescaled centroids between the new base case and the other 12 cases at different stages of head recovery were computed and plotted (Figure 9).

For the heterogeneous cases, if long-term pumping was performed in a coastal area with large drawdown, corresponding to 0% head recovery in Figure 9, the distances of the rescaled centroids suggest that the TZ was most sensitive to anisotropy in the spatial structure of the hydraulic-conductivity fields in cases 1–6 (Figure 9a,b). Judging from the simulation results, seawater intrusion is facilitated by the anisotropic heterogeneous structure of hydraulic conductivity. This observation implies that seawater intrusion may be significantly underestimated if an isotropic hydraulic-conductivity structure is used for predictive simulations instead of the actual anisotropy.

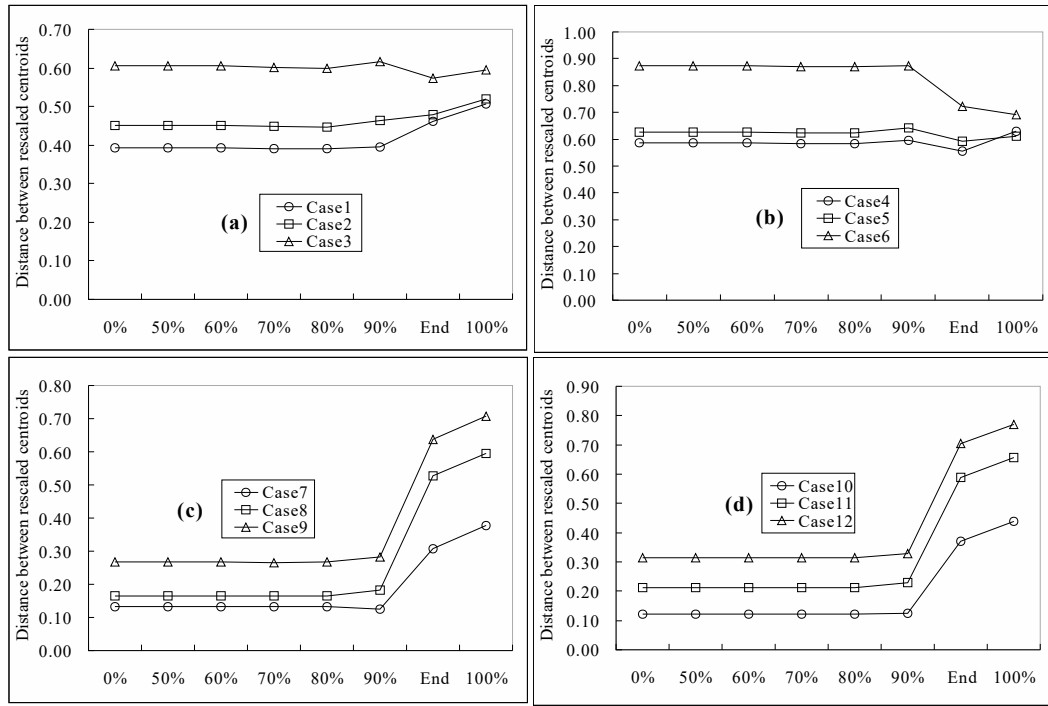

**Figure 9.** Distances of the centroids between the new base case and other cases at different stages of head recovery in the heterogeneous cases.

When the pumping well was shut off, the process of head and TDS recovery began to correspond to greater than 0% head recovery (Figure 9). As head recovery progressed, the trend of the rescaled distances of the TDS centroids did not obviously change from 0% head recovery until the end of the simulation. At the end of the simulation, the distances of the rescaled centroids greatly increased in cases 7–12 (Figure 9c,d) and had similar values as in cases 1–6 (Figure 9a,b). This result suggests that the difference between the sensitivity of the TZ to isotropic and anisotropic spatial structures of hydraulic-conductivity fields had vanished. This observation remained true even when both the head and TDS had completely recovered, corresponding to 100% head recovery in Figure 9.

Increases in the correlation length greatly increased the rescaled distances of the TDS centroids at the end of the simulation and 100% head recovery in cases 7–9 (Figure 9c) and 10–12 (Figure 9d). This result suggests that the sensitivity of the TZ to the correlation length of the hydraulic-conductivity field greatly increased when the head recovery was almost completed for the cases that considered an isotropic spatial structure for the hydraulic conductivity field. However, this increased variance did not significantly affect the distances of the centroids in cases 7–9 (Figure 9c) and 10–12 (Figure 9d) when the head recovery had been completed. Therefore, the TZ is less sensitive to the variance of the hydraulic-conductivity field.

The distances of the rescaled centroids in cases 4–6 (Figure 9b) were larger than those in cases 1–3 (Figure 9a), suggesting that the variance of the hydraulic-conductivity field is more important than the correlation length in controlling the TZ's location when an anisotropic spatial structure is considered for the hydraulic-conductivity field. The distances of the rescaled centroids increased when the correlation length increased in cases 1–3 (Figure 9a) and cases 4–6 (Figure 9b), suggesting that larger correlation lengths increase the effects on the TZ's location.

To understand the recovery of TDS with time, the distances between different stages of head recovery were computed for the same cases, and the accumulated distances from the beginning of the recovery to different stages of head recovery were then derived. Figure 10 shows the accumulated distances of the rescaled centroids for different cases.

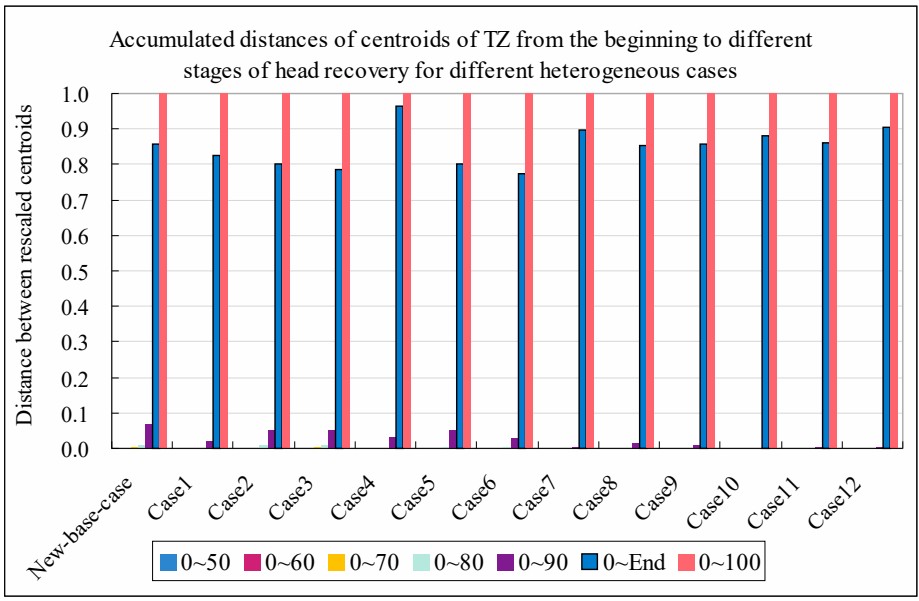

**Figure 10.** Accumulated distances of the rescaled centroids from the beginning of recovery to different stages of head recovery for different heterogeneous cases (note: "New base case" in the figure denotes the new base case).

When the head recovery was less than 90%, the accumulated distances for the different cases were close to zero, implying that no significant TDS recovery occurred until 90% head recovery was reached for all the simulated cases. When the percentage of head recovery was equal to 90%, the accumulated distances of the 12 cases were generally less than 10%. The distances in cases 7–12 were especially low. This result suggests that the process of TDS recovery is much slower than that of head recovery. At the end of the simulation, when head recovery was almost completed, the percentage of TDS recovery significantly increased and the absolute value was generally from 80% to 90%, indicating that a large amount of TDS recovery occurs only when the head approaches the original state without pumping. Figure 10 shows that the TDS recovery in cases 7–12 was much faster than that in cases 1–6, except for case 4.

## 5. Conclusions

The recovery of head and the TDS as a result of long-term pumping and deactivation in a 2D coastal aquifer system was studied with a numerical method. The first spatial moment was chosen to measure the centroid of the TZ between freshwater and saltwater. Twelve homogenous cases were designed to study the effects of different parameters on the TZ's location for both homogeneous and heterogeneous aquifers, and sensitivity analysis was conducted to evaluate the sensitivity of the TZ to different hydrological parameters.

In the homogeneous cases, the recovery of head and the TDS had different behaviors after long-term pumping in coastal aquifers. Head recovery was achieved within a short time compared to TDS recovery. Generally, fewer than 10 years were required to reach 90% head recovery for the designed cases in this study. However, the TZ did not show substantial movements within the first 10 years and the recovery was still incomplete even after 100 years of simulation. Sensitivity analysis showed that the hydraulic conductivity and rainfall recharge were the two most sensitive factors that affected the location of the TZ during the recovery process. High hydraulic conductivity and low recharge rate may cause the landward movement of the TZ. The anisotropy of the aquifer was another important factor that affected the TZ's movement (i.e., landward movement for an anisotropy ratio > 1 and seaward movement for the opposite case). The dispersivity only affected the location of the TZ when the head recovery was less than or equal to 90%, and the sensitivity of the TZ's location to

the dispersivity was negligible compared to the other parameters, especially when the TDS were fully recovered. Small porosity and large specific storage played an important role in the location of the TZ compared to the other parameters (except for the hydraulic conductivity and recharge rate) when 90% of the head was recovered. The porosity became much more important compared to the dispersivity and specific storage at the late TDS-recovery stage.

In the heterogeneous cases, the hydraulic-conductivity field was assumed to be a log-normally distributed correlated random field. The property distributions of the aquifer, such as isotropic and anisotropic spatial structures, the variance, and the correlation length, were designed by a trial-and-error method to avoid the numerical-convergence problem. The required time for complete head recovery was much shorter than that in the homogeneous cases, e.g., 90% head recovery within 400 days. However, the TDS recovery was negligible when 90% of the head was recovered, and significant TDS recovery occurred between 90% head recovery and the end of the simulation. Complete TDS recovery was not achieved even after 100 years of simulation. The sensitivity analysis suggested that the sensitivity of the TZ to the correlation length and variance parameters largely depended on the spatial structure of the hydraulic-conductivity field. If an isotropic spatial structure was considered, the TZ was more sensitive to the correlation length than the variance. The sensitivity of the TZ to both the correlation length and variance greatly increased when the head recovery was nearly complete. If an anisotropic spatial structure was considered, the TZ was more sensitive to the variance than the correlation length throughout the recovery process. When the variance and correlation length were large, the TZ could move further landward. The random log-normal hydraulic-conductivity distribution caused the TZ to move towards the land in contrast to the homogeneous cases, suggesting that a simplified homogeneous model may underestimate the location of the TZ during seawater-intrusion evaluation for a real coastal aquifer.

The 2D models that were applied in this study were highly simplified compared to real aquifer systems and therefore have limitations. For example, the synthetic aquifer that was considered in the study was unconfined, while confined or multi-layered aquifers could be present in real coastal aquifer systems. In such cases, the results from this study should be carefully applied. Stochastic parameters, such as extremely high/low values of hydraulic conductivity, were also constrained by the convergence of the model. Although the hydraulic properties that were used in the case studies were determined by referencing values from an actual coastal aquifer, particular conditions and broader values should be tested to increase the practicality of this study. The screen depth and the location of the pumping well may affect the simulation results. The 2D model itself imposes limitations such as the confinement of the transverse flow and mass transfer.

Nevertheless, the results of this study are meaningful for coastal-aquifer management and may be instructive in the restoration of coastal areas that are experiencing seawater intrusion because of the long-term overexploitation of fresh groundwater. When any evaluation or modeling work on seawater intrusion or TZ movement in a real coastal aquifer is undertaken, the permeability and other sensitive parameters as delineated in this study should be carefully evaluated to avoid any underestimation of the balance between freshwater and saltwater. Understanding the characteristics of head and TDS recovery in a real coastal aquifer can also help optimize the determination of the number of extraction wells and the corresponding extraction rates during aquifer restoration from seawater intrusion.

**Author Contributions:** Conceptualization, E.P. and G.D.; Methodology, H.C.; Software, H.C.; Validation, G.D. and C.H.; Formal Analysis, H.C.; Investigation, Y.K.; Resources, E.P.; Data Curation, J.J.; Writing-Original Draft Preparation, H.C.; Writing-Review & Editing, J.J.; Visualization, C.H.; Supervision, E.P.; Project Administration, Y.K.; Funding Acquisition, E.P.

**Funding:** We would like to express our gratitude to the National Natural Science Foundation of China (41401539) for supporting this study. Financial support for this study was also provided by the Korea Environmental Industry and Technology Institute (project title: Development and field verification of environmental risk estimation system for $CO_2$ leakage, project 2018001810004).

**Acknowledgments:** We give special thanks to the anonymous editors and reviewers for their valuable comments, which improved this manuscript's quality.

**Conflicts of Interest:** The authors declare no conflict of interest.

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
