# Peer review of "A Study on the Recovery of Head and the Total Dissolved Solids (TDS) from Long-Term Pressure Depressions in Low Permeable Coastal Aquifers"

_water, doi:10.3390/w11040777_

Round 1

Reviewer 1 Report

Very interesting work. I would very much like to see this implementation in a real case scenario with actual data.

Author Response

Response: Thank you for the positive comments and practical suggestion. We are also eager to continue our work on the implementation of this study’s results in a real case with actual data.

Reviewer 2 Report

This manuscript investigates the recovery of head and TDS in a coastal aquifer system in terms of long term behaviour. More specifically, the spatial behavior of the Transition Zone (TZ) was chosen as an indicator. Different scenario were modelled with a 2D coupled density-driven flow and solute transport model. All parameters were hypothesized.

The topic is interesting. However I think that the main critical issue is connected with the value of hydraulic conductivity  used in all scenarios.

Indeed, as reported in Tab. 1, the range of hydraulic conductivity  is between 2.5×10-7 m/s and 6×10-7 m/s. In the conclusion the authors write that ….”Hydraulic conductivity and rainfall recharge were the two most sensitive factors affecting the  location of TZ for the whole recovery process….”

Thus, why to create 12 scenarios with a hydraulic conductivity covering a relatively narrow range of values? Moreover, the used hydraulic conductivity is very low.

I think that a more complete analysis should consider a broader range of hydraulic conductivity.

Title

I don’t like the combination of the words “Hypothetica”l and “study”. It is not an Hypothetical Study but it is a study on a hypothetical recovery…..

References

Please, integrate the bibliografy with international references.

Tables

Tab 1: why the simulated hydraulic conductivity is so low? Please, explain in the text the reason of your choice.

Author Response

This manuscript investigates the recovery of head and TDS in a coastal aquifer system in terms of long term behaviour. More specifically, the spatial behavior of the Transition Zone (TZ) was chosen as an indicator. Different scenario were modelled with a 2D coupled density-driven flow and solute transport model. All parameters were hypothesized.

The topic is interesting. However I think that the main critical issue is connected with the value of hydraulic conductivity used in all scenarios.

Indeed, as reported in Tab. 1, the range of hydraulic conductivity  is between 2.5×10-7 m/sand 6×10-7m/s. In the conclusion the authors write that ….”Hydraulic conductivity and rainfall recharge were the two most sensitive factors affecting the  location of TZ for the whole recovery process….”

Thus, why to create 12 scenarios with a hydraulic conductivity covering a relatively narrow range of values? Moreover, the used hydraulic conductivity is very low.

I think that a more complete analysis should consider a broader range of hydraulic conductivity.

Response: We admit that a broader range for the hydraulic conductivity is necessary for more practical usage of this study’s findings. However, the recovery of head and the recession of the transition zone in a coastal aquifer system were targeted in this study with realistic hydraulic properties by referencing the values of an actual site in South Korea. In the revised manuscript, we stated that “The actual values from a coastal aquifer that consists of fractured granite with a subsurface disposal facility of low- to intermediate-level radioactive waste (Gyeongju in South Korea) were referenced when determining the parameters [10]” (lines 95-98). We further incorporated the limitations of this study as follows: “Although the hydraulic properties that were used in the case studies were determined by referencing values from an actual coastal aquifer, particular conditions and broader values should be tested to increase the practicality of this study” (lines 523–525). In a subsequent study, we plan to conduct advanced sensitivity analyses that consider wide-ranged hydraulic properties according to the reviewer’s recommendation.

Title

I don’t like the combination of the words “Hypothetica”l and “study”. It is not an Hypothetical Study but it is a study on a hypothetical recovery…..

Response: We have noted this concern. The title has been changed to “A Study on the Recovery of Head and the Total Dissolved Solids (TDS) from Long-term Pressure Depressions in Coastal Aquifers” according to the reviewer’s recommendation. 

References

Please, integrate the bibliografy with international references.

Response: The reviewer’s recommendation was not specific, and we assumed that the reviewer recommended incorporating more references to support the idea of this study. We added two more references to the reference list:

Park S.C.; Yun S.T.; Chae G.T.; Yoo I.S.; Shin K.S.; Heo C.H.; Lee S.K. Regional hydrochemical study on salinization of coastal aquifers, western coastal area of South Korea. J. Hydrol. 2005, 313, 182-194. (lines 556-557)

Park, J.B.; Jung, H.; Lee, E.Y.; Kim, C.L.; Kim, G.Y.; Kim, K.S.; Koh, Y.K.; Park, K.W.; Cheong, J.H.; Jeong, C.W.; Choi J.S.; Kim, K.D. Wolsong low-and intermediate-level radioactive waste disposal center: progress and challenges. Nucl. Eng. Technol. 2009, 41, 1-16. (lines 565-566)

Tables

Tab 1: why the simulated hydraulic conductivity is so low? Please, explain in the text the reason of your choice.

Response: The reasoning for the low hydraulic conductivity was incorporated as follows: “The actual values from a coastal aquifer that consists of fractured granite with a subsurface disposal facility of low- to intermediate-level radioactive waste (Gyeongju in South Korea) were referenced when determining the parameters [10]” (lines 95-98).

In addition to the above revisions, much of this manuscript was revised based on the recommendations of the reviewer, which are highlighted in the submitted revised manuscript.

Reviewer 3 Report

The manuscript concerns hypothetical study on recovery of head and total dissolved solids from long-term pressure depression in coastal aquifers. The current version of your manuscript is not a scientific communication. Article is really poor prepared, all figures are vague, eg. Line 146. Figure 1. Map showing conceptual model used in the hypothetical 2D study. Line 205. Figure 2. Time required for different percentage of recovery of head, the same situation occur with the rest figure.  The target should be the international reader and at least the method should be presented to be applicable to other situation. Lack of section Material or Data Source. References, tables and figures have to follow instruction for author, and should have the similar type of font as the rest of the text. State clearly what is the aim of the presented study. Article should be carefully edited according to Author’s guidelines. and should have the similar type of font as the rest of the text. Include in the introduction and the conclusion the value added with respect to existing research. Also the introduction should include information about problem, which should  be investigated, as well as reasons for conducting the research. Authors need to rewrite the purpose of the work more clearly. Authors do not explain well, where is the novelty of the distinguished method. How in practice the results of the presented work can be used? This should be discussed in the point concerning discussion of the results. Some information about practical use of the obtained results, both in the section Results and Conclusions should be underlined. The conclusion of this manuscript appear to be weak and do not contribute to the current understanding of the restoration of coastal areas experienced seawater intrusion. The last point of the article contains in fact only the conclusions relating to the researched case study, but there is no more detailed perspective. I recommend to embed the results of a statistical test, which would reinforce the results.

Author Response

The manuscript concerns hypothetical study on recovery of head and total dissolved solids from long-term pressure depression in coastal aquifers. The current version of your manuscript is not a scientific communication. Article is really poor prepared, all figures are vague, eg. Line 146. Figure 1. Map showing conceptual model used in the hypothetical 2D study. Line 205. Figure 2. Time required for different percentage of recovery of head, the same situation occur with the rest figure.  

Response: We admit that the figures in the previous version were of poor quality. All the figures were fully replaced with high-quality figures.

The target should be the international reader and at least the method should be presented to be applicable to other situation. Lack of section Material or Data Source. References, tables and figures have to follow instruction for author, and should have the similar type of font as the rest of the text. 

Response: This study is a hypothetical study, so a Materials/Data Sources section is not applicable. Regarding the format, this manuscript was re-formatted with the same type of font following the “instructions for author”.

State clearly what is the aim of the presented study. Article should be carefully edited according to Author’s guidelines. and should have the similar type of font as the rest of the text. Include in the introduction and the conclusion the value added with respect to existing research. Also the introduction should include information about problem, which should be investigated, as well as reasons for conducting the research. Authors need to rewrite the purpose of the work more clearly. Authors do not explain well, where is the novelty of the distinguished method. How in practice the results of the presented work can be used? This should be discussed in the point concerning discussion of the results. Some information about practical use of the obtained results, both in the section Results and Conclusions should be underlined. The conclusion of this manuscript appear to be weak and do not contribute to the current understanding of the restoration of coastal areas experienced seawater intrusion. The last point of the article contains in fact only the conclusions relating to the researched case study, but there is no more detailed perspective. I recommend to embed the results of a statistical test, which would reinforce the results.

Response: The portion of the introduction regarding the objective of this study was modified according to the reviewer’s recommendation. The revised version explicitly states that no previous studies have systematically addressed the recession of the transition zone with the cessation of long-term pumping: “The decay in increased salinity that is induced by pumping is expected to take a long time, and the characteristics of the recovery process greatly depend on the flow and transport characteristics but have not been systematically addressed in the literature” (lines 66-68). Additionally, the practical usage of this study was clarified as follows: “The results of this study can provide useful information for the control and management of recovery processes in a real coastal aquifer to establish necessary countermeasures” (lines 76-78). In the conclusion section, the practicality of this study was revised for clarity (lines 530-535). Although statistical tests were not conducted here, the limitations of this study were stated in the conclusion section as follows: “Although the hydraulic properties that were used in the case studies were determined by referencing values from an actual coastal aquifer, particular conditions and broader values should be tested to increase the practicality of this study” (lines 523-525). We plan to conduct advanced sensitivity analyses that consider wide-ranging hydraulic properties in a subsequent study according to the reviewer’s recommendation.

In addition to the above revisions, much of this manuscript was revised for further clarity, which is highlighted in the submitted revised manuscript.

Round 2

Reviewer 2 Report

I suggest to introduce the indication that the study is focused on a low permeability costal aquifer also in the title, abstract and introduction. The main reason is that this study is not related to different scenarios (how I can suppose by reading the title) but is connected only to situation with a  particular hydrogeological  condition.

Author Response

By fully reflecting the reviewer's point, the title is changed to "A Study on the Recovery of Head and the Total Dissolved Solids (TDS) from Long-term Pressure Depressions in Low Permeable Coastal Aquifers" (lines 2-4).

In addition, it is specified in the abstract (line 18) and the introduction (line 72) that low permeable coastal aquifer is the main concern of this study.

Reviewer 3 Report

My numerous remarks were included in the revision.

Author Response

The authors appreciate the reviewer's previous comments in the 1st round review that immensely improve the quality of our manuscript.